# Recruiting Immunity for the Fight against Colorectal Cancer: Current Status and Challenges

**DOI:** 10.3390/ijms232213696

**Published:** 2022-11-08

**Authors:** Ensaf M. Al-Hujaily, Batla S. Al-Sowayan, Zeyad Alyousef, Shahab Uddin, Farah Alammari

**Affiliations:** 1Department of Blood and Cancer Research, King Abdullah International Medical Research Center, Riyadh 11481, Saudi Arabia; 2King Saud Bin Abdulaziz University for Health Sciences, Riyadh 14611, Saudi Arabia; 3Department of Surgery, College of Medicine, King Saud bin Abdulaziz University for Health Sciences, Ministry of National Guard Health Affairs, Riyadh 14611, Saudi Arabia; 4Translational Research Institute and Dermatology Institute, Academic Health System, Hamad Medical Corporation, Doha 3050, Qatar; 5Clinical Laboratory Sciences Department, College of Applied Medical Sciences, King Saud bin Abdulaziz University for Health Sciences, Riyadh 14611, Saudi Arabia

**Keywords:** colorectal cancer, adoptive cell therapy, chimeric antigen receptor-T cell (CAR-T cells), immunotherapy, monoclonal antibodies (mAb), tumor-associated antigen (TAA), clinical trials

## Abstract

Cancer immunotherapies have changed the landscape of cancer management and improved the standard treatment protocols used in multiple tumors. This has led to significant improvements in progression-free survival and overall survival rates. In this review article, we provide an insight into the major immunotherapeutic methods that are currently under investigation for colorectal cancer (CRC) and their clinical implementations. We emphasize therapies that are based on monoclonal antibodies (mAbs) and adoptive cell therapy, their mechanisms of action, their advantages, and their potential in combination therapy. We also highlight the clinical trials that have demonstrated both the therapeutic efficacy and the toxicities associated with each method. In addition, we summarize emerging targets that are now being evaluated as potential interventions for CRC. Finally, we discuss current challenges and future direction for the cancer immunotherapy field.

## 1. Introduction

Colorectal cancer (CRC) is one of the most common malignancies in the world. CRC is characterized by low survival rate (5 years, 10%) in advanced metastatic stages [1]. Almost half of CRC patients develop metastasis, making CRC one of the leading causes of cancer-related deaths [2,3]. The conventional prognostic factors and treatment options for CRCs are based on histologic tumor grade (differentiation) and tumor stage (TNM; tumors/nodes/metastases, stages I–IV) [4,5]. In addition to histological features and TNM classifications, metastatic CRC can also be classified based on the molecular profile of the disease. The CRC metastasis can either be: deficient mismatch repair (dMMR); high level of microsatellite instability (MSI-H) or proficient mismatch repair (pMMR); microsatellite stable (MSS) [6]. Although dMMR/MSI-H metastases are poorly differentiated tumors with a higher mutation rate, they tend to have more tumor-infiltrating lymphocytes (TILs) compared with pMMR/MSS CRCs. Accordingly, dMMR/MSI-H tumors are more sensitive to treatments with immune checkpoint inhibitors [6,7,8].

Standard CRC interventions, whether primary or metastatic, consist of laparoscopic surgery coupled with neoadjuvant or adjuvant chemotherapy [9,10,11,12,13,14]. The most common therapeutic agent for CRC is 5-fluorouracil (5-FU), a thymidylate synthase inhibitor, which converts deoxyuridine monophosphate (dUMP) to deoxythymidine monophosphate (dTMP), causing DNA damage [15]. Although 5-FU has shown to be effective in early stages of CRC, the response rates in metastatic CRC drop to 10–15% [6,7]. This points to the urgency of developing new therapeutic strategies for this disease. In fact, patients with aggressive multi-organ metastatic disease may benefit from doublet chemotherapy in combination with targeted therapy [16,17,18]. For instance, it was reported that a combinatorial chemotherapeutic regiment consisting of 5-FU, in combination with oxaliplatin (FOLFOX) or irinotecan (FOLFIRI), has shown response rates of 40–50% [15].

Recently, new methods for cancer treatment have been developed thanks to the discovery of various promising immunotherapeutic techniques [19,20]. Ideally, immunotherapy functions by the production or expansion of patient’s immune response to eradicate tumor cells. This will eventually improve patient’s survival rate while preserving an acceptable toxicity profile [21,22]. Many emerging immune-based techniques have been shown to have a strong impact on tumor eradication. These include monoclonal antibody therapies, cancer vaccines, and adoptive T cell therapy [19,23]. In this review, we highlight the development and the utilization of immunotherapeutic techniques that are based on monoclonal antibodies (mAbs) and adoptive cell therapy. Moreover, we discuss and summarize targets that are newly being adopted and are being evaluated for the treatment of CRC.

## 2. Monoclonal Antibody Therapy

Monoclonal antibody (mAb) therapy has shown promising results in clinical studies for the treatment of CRCs, specifically when used in combination with other therapeutic agents. In order for it to function efficiently, different mechanisms must be utilized in parallel to activate the immune system [24]. For example, mAbs can act as a link between tumor cells and immune effector cells. This works by mAbs binding to the tumor cell antigens through their hypervariable region and the immune cells through their Fc region. As a result, the tumor cells are ultimately destroyed through antibody-dependent cellular cytotoxicity (ADCC) [25]. Another mechanism of mAbs action is known as complement-dependent cytotoxicity [26]. In this mechanism, the mAbs inflict their therapeutic effect by activating proteolytic enzymes and then forming a terminal lytic complex that ruptures the targeted cells’ membrane. An alternative mAbs mechanism, which has more direct effect on tumor cells, is their ability to act as an agonist. Here, mAbs bind and then activate cell surface receptors on tumor cells, which triggers apoptosis. On the other hand, mAbs can also act as antagonists, where they bind to a cell surface receptor and block the downstream signaling pathways that are known to be essential for cell survival and proliferation. In addition, mAbs can act as carriers for therapeutic agents, as they can be paired with radioisotopes, small interfering RNA, or cytotoxic drugs. This will allow for the specific delivery of these agents to their targeted tumor cells, rendering these reagents more efficient compared with when they are systematically infused [27]. Lastly, not all mAbs therapies target tumor cells directly. Instead, mAbs can also display antitumor effects by targeting the host immune cells. In this approach, one specific antibody can be used against several types of cancer. For example, anti PD-1 therapy blocks the cell receptor that is important for immune checkpoint signaling, causing a deterioration in tumor development in many cancer types (Figure 1) [20,28,29]. This strategy, known as immune checkpoint blockade, directly shuts down self-tolerance mechanism, particularly, systems that prevent autoimmunity and are utilized by tumor cells to escape immune response [29,30,31]. Nevertheless, despite its versatile use as an antitumor agent, a major challenge in the development of an effective mAbs therapy is identifying the ideal tumor-specific targets [32]. In CRCs, mAbs targets are extremely diverse and require an extensive discussion to cover [33]. In Section 4 of this review, where we discus emerging CRC targets, we discuss the clinical applications of several mAbs in the treatment of CRC.

## 3. Adoptive Cell Therapy

Adoptive cell therapy (ACT) includes the enhancement, ex vivo expansion, and/or manipulation of autologous and allogeneic immune cells, followed by reintroduction into the patient. The injected cells are expected to migrate to the tumor site and destroy tumor cells. Below, we summarize the main currently used ACT techniques, which involves genetic manipulation, cytokine induction, and vaccine development.

### 3.1. Genetically Redirected Immune Cells

One approach of adoptive cell therapy is to use the patient’s own naturally occurring, tumor-infiltrating T cells to help eliminate the tumor. Despite the effectiveness of this approach, one major challenge that remains is how challenging and expensive it is to isolate and expand these cells in vitro [34]. T cell therapy is also commonly coupled with the administration of the growth factor IL-2 to maximize cells’ in vivo expansion. Moreover, it is often recommended that host lymphodepletion is performed by chemotherapy or together with total-body radiation. This is performed to facilitate homeostatic lymphocytic development and perseverance of the transplanted T cells [34,35]. Therefore, the development of genetically engineered lymphocytes has been a major breakthrough in cancer immunotherapy [36]. As a result, we are seeing an increase in the generation of effector cells that are specific for tumor-associated antigens (TAAs). In this approach, T cells can be engineered to express receptors, such as T cell receptors (TCRs), that are highly specific for a given TAA [37,38]. This occurs by harvesting T cells from the patient, then transgene encoding of the targeted TCR using a viral vector or other carriers in order to target the T cells. When T cells are modified, they are transplanted back into the patients and can identify the target antigen, such as the peptide-major histocompatibility complex (MHC) unit. With all the progress in this application, we still acknowledge some limitations, the most significant being that tumor cells can often reduce the expression of MHC to escape T cell recognition [36,37]. In addition, heterologous pairing with endogenously expressed TCRs can often cause off-target effects that are difficult to predict [39].

Fortunately, there is an alternative MHC-independent approach that has been investigated and studied thoroughly. Chimeric antigen receptor-T lymphocyte (CAR-T) cell therapy is a substitute to adoptive T cell therapy for CRCs [40]. It also requires genetic modification of T cells in order to generate CAR-T cells that present antigen-specific moieties [36]. The chosen effector lymphocytes can also be prepared with improved properties needed for better elimination of tumor cells [34]. In this method, CAR-T cells are made by combining a single-chain variable fragment (scFv) with an intracellular signaling domain (Figure 2). The scFvs are derived from a mAb, specifically targeted for a cell surface TAA. The variable extracellular domain of the antibody will recognize the MHC-independent structure on the target tumor cell surface. When the antigen binds, the intracellular signaling domains will activate the self-renewal and lytic function of T cells. It should be taken into consideration that a major factor for the development of a successful CAR-T cell therapy is identifying the correct antigens. These antigens can be on the tumor cells’ surface or on their permissive microenvironment but are not present on healthy cells [34,36,38]. It is noted that some issues can hinder the clinical application and broad use of CAR-T cell-based therapy, such as the development of ‘on-target, off-tumor’ toxicity, that recognizes antigens in healthy cells. One way of overcoming safety issues in this therapy is the use of cell-fate control or ‘suicide’ elements [41,42,43]. This would cause the depletion of the CAR-T cells if any adverse effects occur [44].

It is important to note that CAR therapy is not limited to T cells. Another CAR-based approach called CAR-NK (natural killer cells) effector cell therapy has been developed. NK cells have certain advantages over T cells. Due to differences in cytokine production compared to T cells, they are less likely to cause cytokine-release syndrome. They also have lower probability to initiate graft versus host disease (GvHD) in an allogeneic setting [45]. Therefore, NK cells may be a safer effector cell population. Moreover, NK cell lines can be used as CAR effector cells, which opens the possibility for an “off-the-shelf” therapy. Yet, these therapies still need to be further investigated. For now, CAR-based therapies have shown promising results in clinical trials that will be discussed in this review. Moreover, aside from genetic editing, Li et al. [46] evaluated the administration of autologous NK cell therapy in combination with conventional chemotherapy in patients with locally advanced CRC. This open-label cohort study shows a significant improvement in 5-year progression-free survival (PFS) and overall survival (OS) rates. Hence, the study points to the application of NK-based cell therapy in combination with chemotherapy in locally advanced CRC to prevent recurrence and to prolong survival.

### 3.2. Cytokine-Induced Killer Cells

A more mixed approach is the cytokine-induced killer (CIK) cells. CIK cells are heterogeneous NK-like T lymphocytes that express both the T cell marker CD3 and the NK-cell marker CD56. They are generated ex vivo by incubation of peripheral blood lymphocytes (PBLs) with interferon-gamma (IFN-g) and monoclonal antibody against CD3 (anti-CD3) and interlukin-2 (IL-2) in a time-sensitive manner [47,48]. Evaluation of CIK cell applications targeting a broad range of tumor tissues showed that CIKs are rapidly proliferative with strong, MHC-independent, cytolytic activity and minimal toxicity [47,48,49]. CIK therapy has been frequently studied in CRC. In particular, combination of adjuvant chemotherapy with infusions of CIK cells significantly increased PFS and OS of CRC patients [50,51]. Furthermore, the same combination was also used on patients with metastatic CRC. Results from this study showed significant improvement in OS with good tolerability [52]. Collectively, these studies indicate that sequential adjuvant CIK cell treatment, combined with chemotherapy, is an effective therapeutic strategy to prevent disease progression and prolong survival of patients with advanced CRC, warranting further evaluation. Indeed, two phase II clinical trials (NCT03329664 and NCT03220984) are still ongoing to evaluate the efficacy of this therapeutic approach in patients with metastatic CRC.

### 3.3. Dendritic Cell-Based Vaccines

A different approach from the ex vivo expansion of anti-CRC effector T cells is the administration of dendritic cell-based vaccines. In this method, dendritic cells are harvested and treated ex vivo to display tumor-specific peptides. Then, the cells are injected back into the patient [53,54]. These antigen-presenting cells (APCs) will cause a strong response by the antitumor T cells [54,55]. This technique has been investigated and assessed in combination with CIK-based therapy for treating patients with advanced CRC, which resulted in an increase in PFS (56), OS [56,57,58], and improved life quality [57] among those patients. This suggests that dendritic cell vaccines together with CIK are a potential effective method for the treatment of advanced CRC.

## 4. Targets for Immunotherapy in CRC

### 4.1. Vascular Endothelia Growth Factor

Angiogenesis is the process of forming networks of blood vessels to enhance growth and survival of normal and cancerous tissues [59]. Many tumor cells secrete the glycoprotein vascular endothelial growth factor-A (VEGF-A), which binds to VEGFR-1 and VEGFR-2. Activation of the VEGF/VEGFR pathway promotes migration and proliferation of endothelial cells and increases vessel permeability. Therefore, VEGF has been an attractive target for cancer treatment, including metastatic CRCs [60,61,62]. Although the mechanism induced by VEGF-targeted therapies still remains to be elucidated, it is thought that VEGF-based therapies were shown to play a role in modulation and activation of the immune response within the TME [63,64]. They repair tumor blood vessel structure through a process known as vessel normalization [61]. For instance, the humanized monoclonal antibody (bevacizumab) targets VEGF-A, which leads to normalizing the leaky tumor vasculature and improving delivery of chemotherapy [65]. This normalization is caused by a VEGF pathway inhibitor, resulting in increased tumor infiltration lymphocytes, such as CD4^+^ and CD8^+^ T cells, into the tumor parenchyma [66]. In agreement, another study showed that VEGF expression levels are associated with decreased CD8^+^ T and TH1 cell response in CRC [67]. Moreover, VEGF signaling can control tumor immunodeficiency by recruitment and activation of inhibitory immune cells, such as regulatory T lymphocytes (Tregs) and myeloid-derived suppressor cells (MDSCs) [68]. Bevacizumab significantly reduces the proportion of Tregs and MDSCs in peripheral blood from cancer patients [69]. Dendritic cells (DCs) and tumor-associated macrophages (TAMs) were also showed to be affected by VEGF-targeting therapies. VEGF can reduce DC precursor cell differentiation into mature cells capable of expressing tumor antigens and enhancing an allogenic T-cell response [70].

However, although bevacizum functions in immunomodulation, it still had a modest effect in treating metastatic CRCs [71]. A phase III clinical trial (AVF2107g) reported improved median OS following treatment with bevacizumab and fluoropyrimidine. In this randomized study, 402 patients with metastatic CRC received irinotecan, 5-fluorouracil, leucovorin, and bevacizumab, while 411 patients received leucovorin alone and a placebo [18]. Median OS for the bevacizumab-receiving group was 20.3 months, while patients who received chemotherapy alone showed OS of 15.6 months; *p* < 0.001. The same study also reported that PFS was 10.6 for patients who received the combination with bevacizumab, while it was 6.2 months for patients who received chemotherapy alone; *p* < 0.001 [18]. Another randomized phase III clinical trial (NO16966) reported that the addition of bevacizumab significantly improved median PFS in patients receiving a combination of fluoropyrimidine and oxaliplatin [72]. Furthermore, the combination of bevacizumab and 5-fluorouracil/oxaliplatin resulted in a significant increase in median PFS, but not median OS, compared to chemotherapy alone in refectory patients with metastatic CRC [71,73]. The addition of bevacizumab along with ex vivo expanded T-cell infusion to a combination therapy, consisting of oxaliplatin and capecitabine, was also investigated. In this particular study, fifteen patients with stage IV CRC achieved an overall response rate (ORR) of 80% with tolerated toxicity [74]. Collectively, these studies have demonstrated improvement in OS, PFS, or ORR following the addition of bevacizumab [71,72,73,74].

Another humanized monoclonal antibody that targets angiogenesis through binding to VEGFR-1 is aflibercept. Aflibercept has been suggested to have a broader anti-vascular effect compared with bevacizumab. This is due to its ability to bind and inhibit the placenta growth factor (PLGF), which stimulates angiogenesis by activation of the VEGFR-1. When used with a combination of irinotecan as a second-line treatment for metastatic CRCs, aflibercept was shown to improve both PFS and OS in a phase III clinical trial (VELOUR trial; NCT00561470) [75]. However, it is worth noting that aflibercept failed to improve patient’s outcomes in first-line treatment when combined with oxaliplatin, with worse toxicities in the aflibercept treatment arm [76]. Furthermore, since VEGF inhibitors have an anti-vascular effect, it is expected that they have vascular complications, such as wound-healing delay, bleeding and thrombosis, hypertension, and proteinuria [77].

### 4.2. The K-RAS/B-RAF Pathway

Accumulation of mutations in oncogenes and tumor suppressor genes plays a critical role in colon carcinogenesis [78]. Tumor suppressor genes, or anti-oncogenes, transduce negative cell growth regulation signals, whereas oncogenes promote cell growth [79]. Once tumor suppressor genes are inactivated, the cell escapes cell cycle control and is predisposed to uncontrolled growth and division. Loss of function of multiple tumor suppressor genes is thought to be the major event leading to the development of human malignancies [80]. Oncogenes with a proven role in colorectal cancer are *RAS*, *EGFR*, and TGF-beta 1 [81]. *KRAS-* or *BRAF*-activating mutations, which are connected to the RAS/MAPK pathway, are found in 32–37% and 10%, respectively, of colorectal tumors [82,83,84]. Mutations in the gene PI3KCA, which codes for the catalytic subunit of the PI3K protein implicated in the PI3K/AKT pathway, were reported in 15% of the colon cancer patients [85,86]. The main tumor suppressor genes that participate in colon cancer are *APC* and *P53*. Patients with mutant forms of the *P53* gene frequently respond poorly to existing treatments [87]. One of the early events in the onset and spread of colorectal cancer is thought to be mutations in the *APC* gene [88] (Figure 3).

The three human *RAS* genes (*K-RAS*, *N-RAS*, and *H-RAS*) are considered to be the most frequently mutated oncogenes in human cancers [89,90]. RAS protein is located in the inner cell membrane and activated upon binding of an extracellular ligand to a receptor tyrosine kinase. RAS activation induces the formation of an RAF-1/B-RAF heterodimer, which induces cell proliferation and differentiation through induction of the mitogen-activated protein kinase (MAPK) and the MAPK/ERK kinase (MEK) signaling pathway [90,91]. Overexpression of K-RAS has been reported in 50% of patients with early stages of CRC [92,93,94]. However, it is widely accepted that *K-RAS* mutation is not the primary initiating event, rather the loss of *APC* [95] or β-catenin mutations in mismatch-repair-deficient tumors [96]. The degree to which CRC progression is dependent on *K-RAS* is still unclear. Nonetheless, the high occurrence of *K-RAS* mutations makes it a meaningful target in the treatment of CRCs [89]. It has been shown that KRAS can induce an immune modulatory effect via several downstream pathways. For example, stimulation of NF-κB activates several chemokines and cytokines, such as TNF-α, IL-1, IL-6, CXCL1, 2, 5, 8, and RAF/MAPK. It can also stimulate cytokines independent of NF-κB, via activation of PI3K, which induces IL-10, transforming growth factor β (TGF-β) and granulocyte–macrophage colony-stimulating factor (GM-CSF) [97,98]. Moreover, several reports have confirmed a link between oncogenic KRAS and the expression of the programmed death receptor-1 (PD-1) in cancer, which is an important molecule to target to avoid resistance to immunotherapy. *KRAS* mutations lead to induced *PD-L1* expression and increased CD8^+^ tumor-infiltrating lymphocytes, reducing tumor-specific T cell functions, which are correlated with an inflammatory TME [99,100]. An alternative mechanism in which KRAS can enhance immunosuppression in cancer is through the stimulation of Tregs in the TME. This was induced via the activation of MEK/ERK/AP-1 by the secretion of IL-10 and TGF-β1 [101]. KRAS mutation can also cause a reduction in the major histocompatibility complex (MHC) class I molecules, which negatively affects CD8^+^ cytotoxic T cell ability to identify cancer cells [102]. In agreement, one study has mutated *KRAS* in a poorly immunogenic CRC cell line that resulted in stimulation of immunity and cancer regression due to the secretion of the cytokine, IL-18 [103]. Furthermore, in CRCs, *KRAS* mutation can induce GM-CSF in the TME through enhancing the infiltration of MDSCs, causing a reduction in anti-tumor immunity [104]. This confirms another role of the oncogenic KRAS; in addition to its pro-tumorigenic effect, it stimulates the recruitment of specific immune cells, which results in immune escape. Importantly, KRAS can cause immunosuppression in CRC, allowing for tumor progression through inhibiting interferon regulatory factor 2 (IRF2), which results in an increase in the expression of MDSCs, supporting their migration to TME [105].

In addition, it has been reported that *B-RAF* mutations occur in approximately 8–12% of CRC patients [106,107], with the worst prognosis associated with the V600 mutation in particular [108]. Similar to *K-RAS*, *B-RAF* mutations induce MAPK signaling and have similar effects on colorectal tumorigenesis [109,110,111]. It has been reported that patients with CRC may carry either a *K-RAS* mutation or a *B-RAF* mutation, as no CRC patients were reported to carry both mutations [112]. Oncogenic BRAF can stimulate changes that allow for immune escape of cancer cells via activation of MAPK/ERK signaling. This is associated with the production of immunosuppressive cytokines, such as IL-6, IL-10, and downregulation of MHC I by cancer cells, which reduces the maturation of dendritic cell (DC) and lower Treg recruitment into the tumor microenvironment (TME) [113,114,115]. Moreover, it has been shown that knockdown of *BRAF* causes a significant immunological effect in the TME that includes enhanced production of IFN-g and TNF-a, tumor-infiltrating lymphocytes (TILs) [116], increased cytotoxic T cells [116], and higher expression of MHC on cancer cells [117], and lower production of immunosuppressant cytokines, such as IL-6, IL-10 and VEGF [117], which are in favor of anti-tumor immunity.

Several therapeutic agents have been developed against K-RAS and B-RAF with minimal impact if used as monotherapy [118]. For instance, vemurafenib, an oral inhibitor of BRAF V600 kinase, has limited success in metastatic CRC when used as a single agent. The reasons behind this are still unclear, but in vitro analysis has suggested that mechanisms of resistance to B-RAF treatment may include EGFR over-activation [119] and activation of the PI3K/AKT pathway [120]. Accordingly, current strategies use combinations of targeted inhibition as treatment for BRAF V600E-mutated metastatic CRC instead of a single-agent-based therapy. For example, the phase II trial (SWOG S1406; NCT02164916) established a significant increase in PFS following the addition of vemurafenib to the combination of irinotecan and cetuximab (EGFR inhibitor). This a second line and beyond therapy for patients carrying the BRAF V600E mutation [121]. Additionally, in the ongoing phase III BEACON CRC trial, 30 CRC patients with BRAF V600E mutation received a combination of the B-RAF inhibitor (encorafenib), with an inhibitor of its downstream target MEK (binimetinib), in addition to cetuximab. The study confirmed an ORR of 41% with tolerated toxicities [122], providing additional evidence of the meaningful clinical activity of this regimen. Of note, the study also reported side effects that were consistent with known BRAF, MEK, and EGFR inhibitor toxicities. The BRAF inhibitors’ adverse events include fatigue, rash, diarrhea, pulmonary toxicities, and ophthalmic changes, when combined with MEK inhibitors [123].

### 4.3. Epidermal Growth Factor Receptor

Epidermal growth factor receptor (EGFR) is a tyrosine kinase receptor that, when phosphorylated, induces cell proliferation, migration, survival, and angiogenesis [124]. It is overexpressed in approximately 80% of all CRCs and its overexpression correlates with reduced survival and increased risk of metastases [125,126]. Accordingly, EGFR serves as a meaningful target in the treatment of CRC and its metastases. The EGFR signaling can be blocked by mAb specific to the extracellular domain of the receptor. This will inhibit the receptor dimerization or small molecules that fit into the ATP-binding pocket of the receptor cytoplasmic tyrosine kinase domain [124]. Most clinical data on CRCs are available for receptor-blocking antibodies, such as cetuximab and panitumumab [127,128]. Cetuximab is a chimeric murine human IgG1 mAb, while panitumumab is a humanized IgG2 mAb. In addition to its abilities to inhibit the oncogenic EGFR pathway signaling, cetuximab induces an antibody-dependent cell-mediated cytotoxicity when bound to NK cells [129]. Although in terms of mechanism of action, cetuximab seems to be superior to panitumumab, cetuximab has a higher risk of inducing hypersensitivity reactions compared with panitumumab [130,131]. This fact might serve as an advantage of panitumumab over cetuximab, since both drugs are FDA approved for metastatic CRC. Both antibodies show efficacy in chemotherapy-naive patients as well as in patients whose tumors are refractory to chemotherapy by improving the ORR. When combined with chemotherapies, these mAbs also improve PFS and even OS in patients with metastatic CRCs. In addition, the ORRs improved for patients with CRC when treated with cetuximab alone or in combination with irinotecan [132,133]. In the phase III clinical trial (CRYSTAL; NCT00154102), 599 CRC patients received irinotecan, either alone or in combination with cetuximab as a first-line therapy. This study showed that the median OS in the irinotecan/cetuximab and irinotecan-alone groups was 24.9 and 21.0 months, respectively, in the K-RAS wild-type population. However, median OS in the irinotecan/cetuximab and irinotecan-alone groups was 17.5 and 17.7 months, respectively, in the *K-RAS*-mutant population [132]. This dependence of efficacy on *K-RAS* mutational status comes as no surprise, since it is widely accepted that *K-RAS* mutational status can predict non-responsiveness to EGFR inhibitors [134,135]. This obstacle emphasizes the need for research into the efficacy of EGFR-targeted agents in the treatment of CRCs. Since the clinical efficacy of anti-EGFR antibodies is hampered by mutations in *RAS* gene, the application of cetuximab in combination with NK-cell therapy has been investigated in patients with K-RAS and B-RAF. In vitro analysis by Veluchamy et al. [129] provided a rationale to enhance cetuximab efficacy through a combination with NK-cell therapy for metastatic CRC patients, harboring *K-RAS* and *B-RAF* mutation. Likewise, cetuximab has been shown to augment the ADCC antitumor activity of NK-cell therapy against CRC with an *EGFR* overexpression. Hence, the combination of cetuximab and NK cells may be a potential immunotherapy for metastatic CRC patients with increased EGFR expression [136,137]. Pre-clinical studies have also been reported to establish the efficacy of CAR-T cells against the EGFR variant III (EGFR vIII), which seems to be exclusively expressed in malignant cells [137,138]. Haung et al. [139] studied the efficacy of EGFR vIII-CAR- T cells in combination with miR-153, a molecule that inhibits indoleamine 2,3-dioxygenase 1 (IDO1), which is highly expressed and inversely related to CRC patients’ survival. Using an animal model, administration of this combination resulted in complete tumor elimination. In a recent phase I/II trial (NCT03542799), the efficacy and safety of targeting EGFR IL-12 using CAR-T cell therapy as a treatment of metastatic CRC are being evaluated and the results are still pending.

### 4.4. Human Epidermal Growth Factor Receptor

The human epidermal growth factor receptor 2 (HER2 or ERBB2) is a receptor tyrosine kinase, which is over-expressed in several types of human carcinomas [140]. Homodimerization or heterodimerization of HER2 following ligand binding leads to transphosphorylation of its intracellular component. This results in activation of MAPK and phosphatidylinositol 3-kinase pathways (PI3Ks). Once activated, both pathways trigger cell proliferation and survival [141,142]. Amplification of *HER2* is observed in 2% to 11% of metastatic CRC cases and it is thought to contribute to the resistance to EGFR-targeted therapy [143]. Trastuzumab is an anti-HER2 antibody that was the first anti-cancer monoclonal antibody developed [144]. The phase II HERACLES study evaluated the efficacy of trastuzumab and lapatinib as a treatment for patients with metastatic CRC, with wild-type K-RAS but HER2 positive. The recruited patients who were refractory to standard of care, including cetuximab or panitumumab, showed ORR of 30% following treatment with trastuzumab and lapatinib [145]. A more promising ORR of 52% was shown in MyPathway, a multicenter, non-randomized, phase IIa multiple-basket study. In this study, patients with advanced refractory solid tumors harboring mutations in HER2 were treated with pertuzumab plus trastuzumab [146]. Although both studies establish the clinical benefit of HER2 inhibition in refractory metastatic CRC, its role in the overall treatment paradigm remains unclear. Interestingly, similar to VEGF-targeted therapies, wild-type K-RAS seems to be a pre-requisite for successful anti-HER2 therapies [147].

For adoptive cell therapy application, the efficacy of HER-2-targeted CAR-T cell therapy has been confirmed using a CRC xenograft animal model. The study shows tumor regression and elimination with significant survival advantage for animals receiving CAR-T cell infusion, compared with their corresponding control group [148]. Although this study demonstrated that CAR-T-cell therapy may be a promising approach for CRC clearance, one of the major challenges in CAR-T cell therapy application is their on-target/off-tumor toxicities [149]. It was reported that one patient with metastatic CRC to the liver and lung died following treatment with HER2-specific CAR-T cells. This might be due to CAR-T cells’ response to low HER2 levels on the lungs’ epithelial cells [150]. As a result of this unfortunate event, adverse events following CAR-T cell therapy for solid tumors have received extensive attention. With that being said, it is worth noting that an ongoing clinical trial (NCT04727151) continues to explore the potential role of T-cells expressing T-cell Antigen Coupler (TAC) that targets HER2 in solid tumors [149,151].

### 4.5. Immune Checkpoint Inhibitors

The anti-programmed death-1 (PD-1), anti-programmed cell death ligand 1 (PD-L1), and anti-cytotoxic T-lymphocyte-associated antigen-4 (CTLA-4) are inhibitory immune checkpoint molecules that are required to maintain self-tolerance. Similarly, upregulation of these molecules allows tumors to escape immune surveillance [152,153]. The PD-1 receptor acts as a dominant negative regulator of antitumor T-cell effector, by engaging PD-L1, which is stimulated by inflammatory cytokines [154]. CTLA-4 is exclusively expressed on T cells and acts as a negative regulator of the initial priming of T cells, as it outcompetes CD28 in binding to costimulatory molecules (CD80 and CD86) located on the APCs (121). As mentioned above, TCR recognition of TAAs presented on the surface of APCs leads to T-cell activation. However, triggering TCRs can enhance *PD-1* expression and, subsequently, *PD-L1* expression by targeted tumor cells, which turns off antitumor T-cell responses, a phenomenon known as T-cell exhaustion [155]. This phenomenon provides the basis for the advances seen in the use of immunotherapy in cancer, since inhibition of PD-1/PD-L1 signaling is an attractive target for cancer immunotherapy. 

Pembrolizumab and nivolumab are humanized mAbs that bind to PD-1, inhibiting its engagement with PD-L1, while atezolizumab is humanized mAb that directly inhibits PD-L1. Application of these antibodies leads to disruption of immune escape and activation of cytotoxic T-cells in refractory patients with solid tumors [62,156]. However, this is not the case for metastatic CRC, as their benefits are restricted to the 3–7% of patients with dMMR/MSI-H [157]. Both Pembrolizumab and nivolumab are FDA-approved agents for subsequent-line treatment of metastatic CRC patients with dMMR/MSI-H. The phase II clinical trial (KEYNOTE-164; NCT 02460198) evaluated pembrolizumab in 32 patients with metastatic CRC (11 with dMMR and 21 with pMMR), who are heavily pretreated with three or four therapies [158]. The study demonstrated an ORR of 40% and of 0% in the dMM and pMMR arms, respectively. This study reports a robust and durable clinical benefit of PD-1 inhibition by pembrolizumab, suggesting its application in first-line treatment of dMMR/MSI-H metastatic CRC. This was taken into action in the phase III clinical trial (KEYNOTE-177; NCT02563002), which evaluates the clinical benefits of pembrolizumab versus 5-fluorouracil–based chemotherapy. This trial included 307 patients with dMMR/MSI-H metastatic CRC who had not previously received treatment. Treatment with pembrolizumab resulted in a doubling of PFS, compared with that of chemotherapy alone (median = 16.5 months vs. 8.2 months; *p* = 0.0002). While a complete response of 11% has been achieved in the pembrolizumab group, it was only 4% in the chemotherapy group [159]. Although this study demonstrated durable clinical benefits of pembrolizumab in dMMR/MSI-H metastatic CRC patients, 30% of patients treated with pembrolizumab had primary resistance [159], emphasizing the need for additional immunotherapeutic regimens. Dual inhibitors of immune checkpoint inhibitors by inhibiting the PD-1/PD-L1 pathway and the CTLA-4 pathway have been investigated in the phase II clinical trial (CheckMate 142; NCT02060188). This trial studied the efficacy of nivolumab alone or in combination with ipilimumab (CTLA-4 inhibitor), in 74 metastatic CRC patients with dMMR who were previously treated. Treatment with nivolumab alone resulted in an ORR of 31.1%, a PFS rate of 50%, and an OS rate of 73% at 1 year. Addition of ipilimumab demonstrated even higher ORR, PFS, and OS rates of 55%, 71%, and 85%, respectively, at 1 year [62,160,161]. When a subset of the recruited patients (45 patients) received the combination as the first-line treatment, ORR and OS rates of 60% and 83%, respectively, were achieved at 1 year, reporting promising preliminary results [160]. Furthermore, an ongoing phase III clinical trial (CheckMate 8HW; NCT04008030) was designed to evaluate the efficacy of nivolumab monotherapy, nivolumab plus ipilimumab, or chemotherapy for patients with dMMR/MSI-H metastatic CRC [162]. 

With all the durable clinical efficacy achieved by targeting immune checkpoint inhibitors, it is worth mentioning that the reported toxicities of these agents are largely due to T-cell activation against self-tissue with subsequent autoimmune response, including colitis and hepatitis [163]. Moreover, engineering T cells with a PD-1 inhibitor has been reported to achieve a durable response in both hematological and solid tumors [164]. Additionally, when CAR-T cells co-express a PD-1 decoy receptor, which replaces the PD-1 with the costimulatory domain of CD28 or IL-7 receptor, they exhibit more persistent antitumor activity against various solid tumors [165,166]. These PD-1-targeting strategies have opened the field for further development of T-cell therapy and extensive work is needed for implementing these methods in CRC.

### 4.6. Carcinoembryonic Antigen

The carcinoembryonic antigen (CEA) is a cell-adhesion glycoprotein that is currently the most common tumor marker in CRC. Normally, CEA expression plays an important role during development but is almost undetectable in normal adult tissues, except in the gastrointestinal tract at a low level [167]. CEA is reported to be overexpressed in several types of solid tumors, including cancers of the gastrointestinal tract, breast, lung, ovary, and pancreas [167,168]. Its expression levels on the surface of adenocarcinoma cells correlate with their increased metastatic potential [167]. Increased expression of CEA on tumor cells promotes uncontrolled proliferation and invasion, due to its interaction with transforming growth factor-β (TGF-β) receptor 1 (TGF-β R1) and, thus, disruption of the TGF-β signaling [169]. Further, CEA has anti-apoptotic actions, through activation of the PI3K survival pathway and inactivation of caspase-9 and caspase-8 [170]. In fact, CEA is a tumor marker and a prognostic factor in colorectal cancers. Serum CEA levels are elevated in 50% of patients who have metastasized tumors to the lymph nodes and in 75% of patients with distant metastases [171]. The expression of CEA on the surface of disseminated colorectal cancer cells isolated from intra-peritoneal lavage is elevated and associated with advanced tumor stages and poor prognosis [172]. This is consistent with the hypothesis that CEA may enhance metastasis by functioning as an adhesion factor that facilitates homing of disseminated tumor cells in the new niche [173].

The radio-labeled monoclonal antibody (^131^I-labetuzumab) was tested in a phase II clinical trial in 23 patients with liver metastasis of CRC who have already gone under tumor resection. The study provided evidence of a promising survival advantage of the adjuvant immunotherapy after long-term follow-up [174]. A more recent study reported that repeated infusion of ^131^I-labetuzumab is feasible but is associated with hepatotoxicity in 63 patients, following complete tumor resection for liver metastasis of CRC [175].

CEA is also one of the most studied targets for anti-CRC CAR-T cells. In vitro and in vivo analysis demonstrated an antitumor activity of CEA-targeted CAR-T cells against CRC, which was significantly enhanced by the addition of IL-12 [176]. The idea of using immune-modulatory proteins in addition to CEA-directed CAR-T cells was also investigated by Hombach et al. [177]. The study used mesenchymal stem cells (MSCs) as a delivery vehicle for IL-7 and IL-12 cytokines, in combination with CEA-specific CAR-T cells. MSCs releasing IL7 and IL12 were superior to non-modified MSCs in terms of mediating the antitumor toxicity induced by CEA-CAR T cells in a transplant tumor mice model. Using CRC tissues from metastatic CRC patients that had survived previous conventional chemotherapy, Osada et al. [178] determined that T-cell-induced toxicity is mediated by CEA/CD3-bispecific T-cell-engaging BiTE antibody. As reported in other studies, these bispecific molecules might activate T-cell-induced lysis of tumor cells, representing a potential treatment option for patients with CEA-positive tumors [178,179,180]. Similar results were achieved using dual CAR-T cells, targeting both CEA and CD30. When compared with CEA-CAR-T cells alone, the infusion of dual CAR-T cells showed significantly enhanced toxicity against established CEA-positive and CD30-negative CRC tumors in a mouse model [181]. Furthermore, the application of CEA-targeted CAR-T cells has been evaluated on mice who received CEA-specific CAR-T cells, with blockage of inhibitory immune molecules, including myeloid-derived suppressor cell (MDSC) depletion and GM-CSF neutralization. Since metastatic spread of CRC to the peritoneal cavity is common and difficult to treat, the study focused on intraperitoneal delivery of CEA-targeted CAR-T cells as a mode of localized delivery. Localized intraperitoneal infusion of CAR-T cells resulted in superior protection against CEA-expressing peritoneal tumors, along with an increase in effector memory T cells over time, when compared with systemically infused CAR-T cells [182]. Data from these studies support the further development of combinatorial CEA-targeted CAR-T immunotherapy for metastatic CRC.

Parkhurst et al. [183] demonstrated a TCR-based therapeutic strategy where the modified CEA-reactive TCR cells showed an enhancement in tumor recognition in comparison with the wild-type T cells against human CRC cell lines. This study provides a proof of concept that CEA-TCR therapy might serve as a potential candidate for future gene therapy-based trials in the field of CRC immunotherapy. Application of the CEA-TCR therapy was evaluated in a preliminary phase I clinical trial conducted on three patients with treatment-refractory metastatic CRC. Following transfusion with autologous anti-CEA TCRs, a substantial decrease in serum CEA levels (74–99%) and objective tumor regression of liver and lung metastasis in one patient were achieved. Of note, all three patients developed severe transient inflammatory colitis, indicating toxicities of CEA-targeted therapies [184]. Similarly, application of CEA-targeted CAR-T cells was evaluated in a phase I dose-escalation trial. Ten patients who were CEA-positive either showed tumor regression or halted progression. However, the study reported several adverse events, including fever, lymphocyte number decrease, and duodenal perforation, warranting further studies for safety risk [185]. Table 1 summarizes the emerging targets for the management of metastatic CRC, as recently reported by published clinical studies:

## 5. Conclusions

Immunotherapeutic approaches are valuable tools in treating CRC and managing its clinical outcomes. In particular, two techniques that are generating a growing interest are monoclonal antibodies (mAbs) and adoptive cell therapy techniques.

Both techniques have demonstrated effectiveness in clinical trials against CRC. A common issue in all conducted in vivo and clinical studies is identifying the correct target. Identifying the correct targets will allow for a more efficient control of the immune system in a manner that increases the antitumor effect, while, at the same time, reduces adverse side effects. Currently, there are several emerging targets that are being investigated for CRC. Nevertheless, for CRC, tumor-specific targets are known to be extremely diverse, which makes it extremely challenging to identify the optimal ones. Another major challenge for immunotherapy in CRC, and in general, is toxicity. The growing number of clinical investigations in immunotherapy emphasizes the importance of the identification and management of any toxicity that may occur. All in all, there are some limitations that hinder the effectiveness of immunotherapy for CRC. However, research is being directed towards overcoming these limitations, due to the great potential this approach presents for cancer patients.

## Figures and Tables

**Figure 1 ijms-23-13696-f001:**
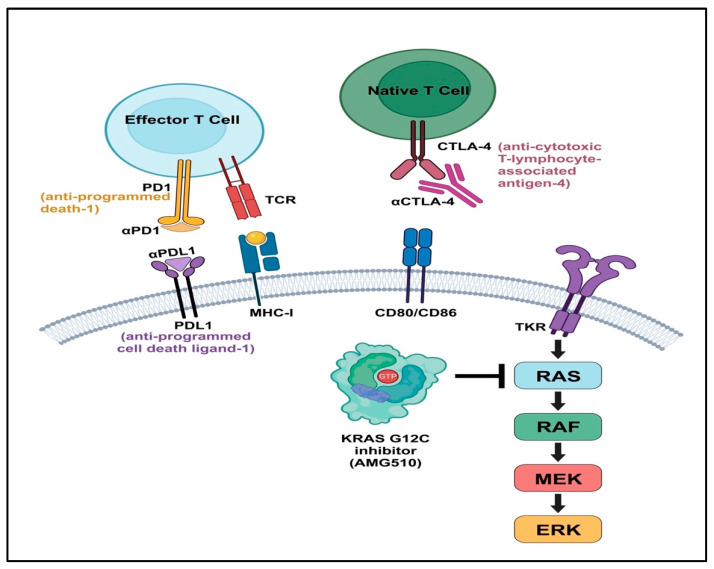
One mAb antitumor mechanism of action is elucidated by directly targeting the host’s immune cells. This schematic illustration demonstrates how PD-1/PDL1 binds and blocks the effector T cell surface receptor that is responsible for immune checkpoint signaling. This halts tumor growth and results in an antitumor effect. PD1: anti-programmed death-1, PDL1: anti-programmed cell death ligand-1, TCR: T cell receptor, MHC-1: major histocompatibility complex-1, CTLA-4: anti-cytotoxic T-lymphocyte-associated antigen-4.

**Figure 2 ijms-23-13696-f002:**
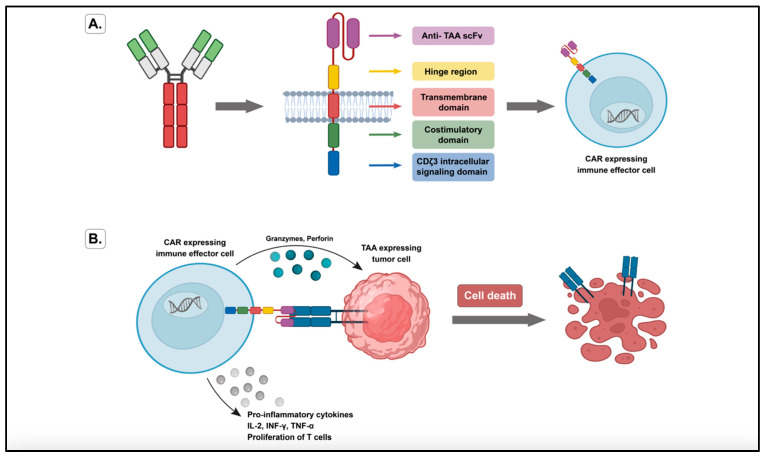
(**A**) Schematic illustration showing the variable light and heavy chains that are sequenced and combined into a single-chain variable fragment (scFv) to convert monoclonal antibody (mAb) to chimeric antigen receptor (CAR). (**B**) Schematic illustration showing the binding between the CAR-expressing immune effector cell and its tumor-associated antigen (TAA)-expressing tumor cell. This binding results in the production of cytotoxic granules and cytokines, which in turn causes stimulation of the host immune system and tumor cell death.

**Figure 3 ijms-23-13696-f003:**
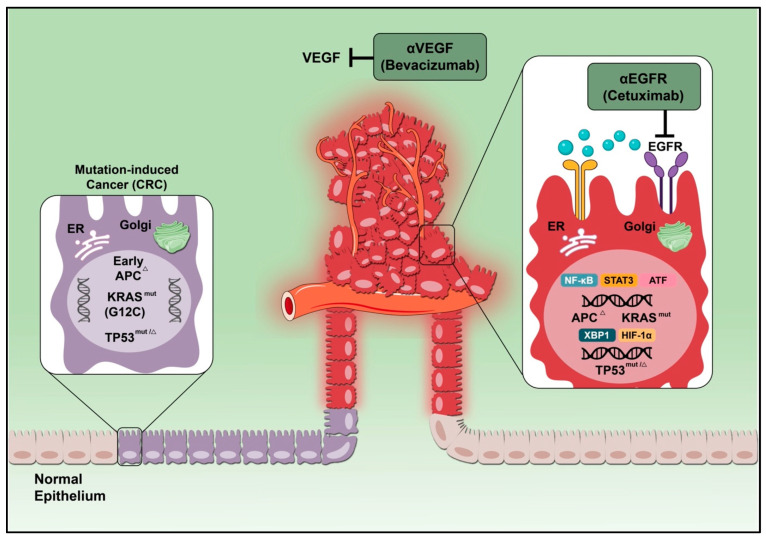
Schematic illustration showing factors that contribute to the development of CRC pathogenesis. Mutation-induced CRC could be a result of new mutations or pre-existing mutations, specifically mutations in tumor suppressor genes and oncogenes. CRC: mutation-induced cancer, VEGF: vascular endothelial growth factor, EGFR: epidermal growth factor receptor.

**Table 1 ijms-23-13696-t001:** Emerging targets for the management of metastatic CRC as recently reported by published clinical studies.

Target	Inhibitor	Results	References
**VEGF**	Bevacizumab	Improved OS, PFS and ORR	[18,71,72,73,74]
	Aflibercept	Improve both OS and PFS	[75]
	Both VEGF inhibitors	Induced vascular-related complications	[77]
**K-RAS/B-RAF**	Vemurafenib	Improved PFS	[121]
	Encorafenib/Binimetinib	Improved ORR, was associated with toxicity	[122]
**EGFR**	Cetuximab	Improved OS	[132]
**HER2**	Trastuzumab/Lapatinib	Improved ORR	[145]
	Pertuzumab/Trastuzuma	Improved ORR	[146]
**PD-1**	Pembrolizumab	Improved PFS	[158,159]
	Nivolumab *	Improved OS, PFS and ORR	[160,161]
**CEA**	^131^I-labetuzumab	Promising survival advantage, associated with hepatotoxicity	[174,175]

* with another inhibitor, CLTA-4 inhibitor, Ipilimumab.

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
