# Peer review of "Recruiting Immunity for the Fight against Colorectal Cancer: Current Status and Challenges"

_ijms, 2022, doi:10.3390/ijms232213696_

Round 1
Reviewer 1 Report
The topic discussed in the study is up-to-date and interesting in the field of colorectal cancer therapy. The paper presents the importance of immunotherapy in the treatment of colorectal cancer.
The first part of the work (Sections 1-3) is written well and is in line with the topic of the work. However, the second part of the work (Section 4) does not coincide with the title of the work.
Major revision should be done in Section 4. Section 4 only mentions possible targets for the treatment of colorectal cancer and hardly anything is written about their importance in immunotherapy. In chapter 6, the importance of these targets (especially VEGF, K-RAS, B-RAF) in immunotherapy in the activation of the immune system should be added, as it is written in the title of the work. For example, it could be noted that using VEGF targeting normalizes the tumor blood vessels thus facilitating the penetration of the immune cells into the tumor and also may polarize the tumor microenvironment to an anti-cancer one as it is written in "Vascular disrupting agents in cancer therapy", doi: 10.1016/j.ejphar.2020.173692.
Some minor corrections should be done:
1. Minor spell check required. For example, verse 95-96 (illistartion - illustration, check point - checkpoint, ereceptor - receptor); verse 161-165 (seaquenced, Scehmatic, illistartion, inturn); vese 339 (trastuzuma - trastuzumab); in Colnclusion: verse 488 (coreect - correct); verse 489 (antituomer); verse 492 (immunetherpy). Some minor language corrections should also be done in the text.
2. Table summarizing the use of different tumor targets in clinical trials can make the text easier to understand.
3. In Section 3.2 Cytokine-Induced Killer Cells more details should be written about the cytokines used, e.g., IL-12, IL-15, IL-2.
4. Figure 4 should be corrected and should described why such factors are essential in immunotherapy of cancer.
Author Response
Thank you for all your comments. Kindly find below our response
Major revision should be done in Section 4. Section 4 only mentions possible targets for the treatment of colorectal cancer and hardly anything is written about their importance in immunotherapy. In chapter 6, the importance of these targets (especially VEGF, K-RAS, B-RAF) in immunotherapy in the activation of the immune system should be added, as it is written in the title of the work. For example, it could be noted that using VEGF targeting normalizes the tumor blood vessels thus facilitating the penetration of the immune cells into the tumor and also may polarize the tumor microenvironment to an anti-cancer one as it is written in "Vascular disrupting agents in cancer therapy", doi: 10.1016/j.ejphar.2020.173692.
Thank you for this great suggestion. the importance of VEGF, K-RAS, B-RAF in immunotherapy in the activation of the immune system has been added to the text
Section 4.1: lines 171-184
Section 4.2: lines 212-222
Section 4.2: lines 232-250
Some minor corrections should be done:
1. Minor spell check required.For example, verse 95-96 (illistartion - illustration, check point - checkpoint, ereceptor - receptor); verse 161-165 (seaquenced, Scehmatic, illistartion, inturn); vese 339 (trastuzuma - trastuzumab); in Colnclusion: verse 488 (coreect - correct); verse 489 (antituomer); verse 492 (immunetherpy). Some minor language corrections should also be done in the text.
Spell check and minor language editing have been done to the text
2. Table summarizing the use of different tumor targets in clinical trialscan make the text easier to understand.
A table summarizing the use of different tumor targets in clinical trials has been added at the end of chapter 4.
3. In Section 3.2 Cytokine-Induced Killer Cellsmore details should be written about the cytokines used, e.g., IL-12, IL-15, IL-2.
In all the references of the studies mentioned in section 3.2, no more details at all were stated about the cytokines used in their studies except what is mentioned here. However, we have included more details about the method of generating cytokine-induced killer cells using peripheral blood lymphocytes (PBLs) with interferon- gamma (IFN-g) and monoclonal antibody against CD3 (anti-CD3) and interlukin-2 (IL-2).
4. Figure 4 should be corrected and should described why such factors are essential in immunotherapy of cancer.
A paragraph about the role of tumour suppressor genes and oncogenes in CRC have been added to the text. In section 4.2 (lines 2012-2022). Moreover, VEGF, KRAS, and BRAF role on immunotherapy has been added to the text.
Reviewer 2 Report
The review article ‘Recruiting Immunity for the Fight Against Colorectal Cancer; Current Status and Challenges’ by Al-Hujaily et al.…., discuss in detail about the Immunotherapy available for Colorectal cancer based on monoclonal antibody and adoptive cell immunity. The authors also provided a detailed discussion of the different limitations and challenges associated with these therapies.
I have the following comments for Author’s consideration-
1. Authors should briefly discuss the role of ‘tumor suppressor genes and oncogenes’ in CRC as presented in figure 3.
2. Authors should mention all the abbreviations (and expansion) in respective figure legends.
3. I also recommend citation of the latest research and review articles on similar topics, for example- Hwang et al., 2021 (biomedicines); Ke Tao Jin et al., 2021 (Cancer Cell International), RM florescu-Å£enea et al., 2019 (Curr Health Sci J) …..etc.
4. Please correct the typos at several places- Line 106…………doptive, Line 163……… illistartion, Line 487……… stides , Line 489……….Adverce, Line 497….. potentioal this apprach…..
Author Response
Thank you for your review and suggestions, kindly find below our response.
1. Authors should briefly discuss the role of ‘tumor suppressor genes and oncogenes’ in CRC as presented in figure 3.
Thank you for the suggestion, a paragraph about the role of tumor suppressor genes and oncogenes in CRC have been added to the text. In section 4.2 (lines 2012-2022)
2. Authors should mention all the abbreviations (and expansion) in respective figure legends.
Abbreviations and expansions in all figure legends have been added.
3. I also recommend citation of the latest research and review articles on similar topics, for example- Hwang et al., 2021 (biomedicines); Ke Tao Jin et al., 2021 (Cancer Cell International), RM florescu-Å£eneaet al., 2019 (Curr Health Sci J) …..etc.
Ke Tao Jin et al., 2021 (Cancer Cell International) and RM florescu-Å£enea et al., 2019 (Curr Health Sci J) have been cited in the text in additions to many references that were published in the past two years.
4. Please correct the typos at several places- Line 106…………doptive, Line 163……… illistartion, Line 487……… stides , Line 489……….Adverce, Line 497….. potentioal this apprach…..
All typos in the text have been corrected.
Round 2
Reviewer 1 Report
The revised version of the article presents the topic from the title of the work much better. Revised Chapter 4 and the description of the importance of these markers in immunotherapy improved the compatibility of the work.
Additionally, the improved figures and the added table increase the readability and consistency of the work.
However, in lines 193-195 in the text "bevacizum" should be corrected to bevacizumab.
After this minor correction, the work can be published in International Journal of Molecular Sciences.